# RHKH: Relational Hypergraph Neural Network for Link Prediction on N-ary Knowledge Hypergraph

## ABSTRACT

All along, KG completion relied on link prediction has always been the focus of researchers. However, overwhelming majority of them can only serve 2-ary KGs. While in practice, knowledge hypergraphs (KH) covering facts beyond binary relations are far more ubiquitous but receive little attention. When confronted with them, massive studies for KGs show inadaptability. The several work towards N-ary KHs generally simply extend KG methods. And they usually transform N-ary knowledge into role-value pairs or triples, largely simplifying inherent association within each piece of knowledge. Furthermore, previous models study each N-ary knowledge independently, resulting in structural correlations among them being completely neglected. Motivated by these, avoiding breaking knowledge structure in KHs like previous studies do, based on original knowledge formats, we propose the first KH reasoning model based on an innovative relational hypergraph neural network (RHNN), RHKH. Challenged by complicated compositions indicated by the original format of N-ary tuples, association within and among each knowledge is discovered through RHNN. It considers complex interactions between relation and entities involved in the same knowledge as well. To refine such interactions, semantic components at each arity-position of relations are distinguished, along with introducing position-specific shift. Extensive experiments demonstrate the effectiveness of our RHKH.

## CCS CONCEPTS

• **Computing methodologies → Knowledge representation and reasoning**; **Machine learning approaches**.

## KEYWORDS

Knowledge Graph, Knowledge Hypergraph, Link Prediction, Hypergraph Neural Network

## 1 INTRODUCTION

Knowledge graphs (KGs) were initially proposed for facilitating information retrieval [40]. With the popularity of KGs, their promoting effect in application scenarios also comes to the fore, including recommendation systems [20], question answering [38] and natural language processing [35]. Up to now, plentiful KGs have been proposed, such as Wikidata [28] and DBpedia [14].

However, even these widely used and recognized KGs are still far from being complete and comprehensive [34]. Such incompleteness seriously affects their value as storage platforms for managing structural information, so as the support for downstream tasks [39]. Motivated by this, KG completion (KGC) becomes a focused research point [5].

KGC is usually realized via link prediction, which is devoted to discovering missing elements in given triples through existing knowledge in the KG. For example, identifying the most possible candidate for $(e, r, ?)$. So far, many studies are targeted for KGC, including TransE [3], ConvE [6] and so on.

Despite such great efforts, overwhelming majority of studies default that KGs only cover instances of binary relations. Knowledge hypergraphs (KHs) with N-ary facts beyond binary relations are significantly neglected. While in practice, N-ary facts occupy a considerable proportion, since most real-world knowledge has inherently complex compositions [7]. Statistically, in Freebase, 61% of the relations are non-binary, covering more than 1/3 of entities [7]. These observations greatly motivate the focus on KHs. An example of comparing KG and KH is in *Figure 1*. As a 3-ary relation *actInFilm*, *Saldana* plays *Neytiri* in *Avatar* is one of its induced instances. For clarity, we call N-ary knowledge in KHs as tuples in the following contents.

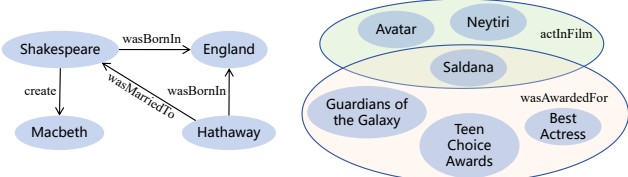

**Figure 1: Knowledge graph V.S. knowledge hypergraph.**

When encountered with KHs, it seems natural to convert its tuples into binary ones and then directly apply KG embedding models. As available methods, reification [4] transforms each N-ary relation into $N$ binary ones and supplement an auxiliary entity to decompose the tuple into $N$ triples. S2C [33] converts the N-ary tuple into $\binom{N}{2}$ triples, each with an auxiliary relation. An example is shown in *Figure 2*. *flight(Brussels, Chicago, Paris)* is converted into $(e_1, flight_1, Brussels)$, $(e_1, flight_2, Chicago)$, $(e_1, flight_3, Paris)$ via reification, and into $(Brussels, flight_{12}, Chicago)$, $(Brussels, flight_{13}, Paris)$, $(Chicago, flight_{23}, Paris)$ via S2C.

However, simplifying KHs into KGs like above fail to yield satisfactory results for link prediction on KHs [7]. Specifically, completing one N-ary tuple requires considering multiple decomposed triples meanwhile. While it is inconsistent with existing KG embedding models since they generally learn and perform link prediction on each triple individually [11]. Abundant auxiliary entities also bring more parameter and efficiency pressure [7]. Furthermore,

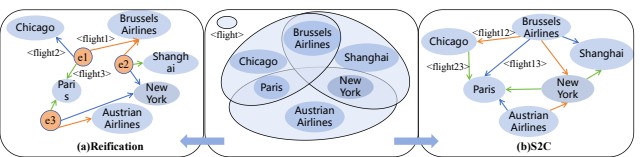

**Figure 2: Convert N-ary knowledge induced by relation *flight* into triples through Reification or S2C.**

this cut associations among entities in the same tuple, causing irreversible loss of structural information [33]. All these potentially affect the accuracy of link prediction. For example, *flight(Brussels, New York, Paris)* is mistakenly regarded as true from *Figure 2(b)*.

Apart from above thought, a few studies propose models targeted for link prediction on KHs as the extensions of studies on KGs. Typically, m-TransH [33] extends TransH [32] by summing projected entities on the hyperplane in each tuple. Meanwhile, they generally reconstruct each N-ary tuple as a primary triple with qualifiers or role-value pairs $\{(r_i : v_i)\}_i^n$ in advance [11, 19]. For example, *actInFilm(Saldana, Neytiri, Avatar)* is reconstructed as *{Actor:Saldana, Role:Neytiri, Film:Avatar}*.

However, their simplifying of internal associations of each tuple cause problems. Firstly, not each N-ary tuple could be considered as possessing a primary triple [11]. Even ignored this, these reconstructions split up each tuple, breaking its complete internal association and relation semantics. Although some work attempts to make up for this, only the likelihood of pairwise co-occurrence of entities, or information flow from role-value pairs to their primary triple are emphasized, rather than the whole associations among involved entities indicated by original knowledge format [37]. Essentially, the methods utilizing primary triples usually still treat link prediction on N-ary tuple as the one on triples, only with additional information $\{(r_i : v_i)\}_i^n$ available [9]. Furthermore, the interaction between entity and relation at different arity-positions in each tuple is not explicitly modeled. In addition to such internal relatedness, association among various tuples is even completely ignored by existing studies. For example, *wasAwardedFor(Chastain, Best Actress, Academy Award, The Eyes of Tammy Faye)* could be the clue for *actInFilm(Chastain, Tammy Faye, The Eyes of Tammy Faye)*.

All these promote us to model tuples based on original knowledge format in KHs, rather than simplifying their inherent complex compositions via reconstruction. Directly confronting complicated interactions in high-ary data naturally poses greater challenge. Association within and among tuples should also be considered. As a powerful tool for discovering associations among entities, GNNs [42] are for graphs, rather than hypergraphs. Typically, for every entity $e$ in the hypergraph, its each neighboring tuple no longer includes only one entity, entities therein also interact with each other when contributing to $e$. Previous hypergraph neural networks [8] do not work in KHs as well since they are all for general hypergraphs, where undirected hyperedges only represent links among nodes, without any meaning or labels, completely different from KHs [7].

Motivated by these, avoiding simplifying or converting knowledge structure in KHs like previous studies do, we innovatively propose a model for link prediction on N-ary KHs based on original

knowledge formats in KHs, RHKH. Particularly, a novel relational hypergraph neural network specifically for KHs is proposed and employed in RHKH. It discoveries correlation within each tuple in KHs through neighboring aggregation. Also, the association among tuples could be established through their co-occurrence in the neighborhood of certain entities. Thus, the key issues mentioned above could be properly answered. Specifically, we adopt a hierarchical aggregation scheme that works within and among tuples successively. The peculiarity of KHs, namely hyperedges are relations with various semantics is highlighted in this process. Furthermore, the entity-relation interaction at various arity-positions in each tuple is modeled as well. And to refine such interactions, for each relation, its semantic components acting on different arity-positions are distinguished. Position-specific offset is also introduced. Contributions of this work are summarized below.

- The usual preprocessing of converting N-ary tuples and picking primary triples in previous studies is omitted in RHKH, possible manual efforts are also saved. Rather than their simplifying or converting of knowledge structure, based on original knowledge formats, we propose the first model for link prediction on KHs based on a novel relational hypergraph neural network (RHNN).
- Challenged by complicated compositions indicated by original knowledge formats in KHs, association within and among tuples are considered through our RHNN.
- Evaluation methods on the credibility of each tuple comprehensively considers its inherent relatedness, including the interaction of involved entities and relations at each arity-position and position-related correlations.
- Extensive experiments fully demonstrate the effectiveness and advantages of our RHKH.

## 2 RELATED WORK

We first outline the representative models proposed for KGs since some of them are the basis for studies on KHs. Then, as the directly related work, studies for N-ary KHs are reviewed. Overall, embeddings methods are the key paradigm for link prediction on KGs and KHs [19]. Generally, they first learn vector representation of entities and relations in respective way. Then, subsequent processes like link prediction are carried out in the obtained embedding space.

### 2.1 Link Prediction on KGs

Most of existing studies are for KGs, divided into translation methods, bilinear models and the ones based on neural network. Translation models are represented by TransE [3], it regards relations as semantic translations from head to tail entity, namely $e + r \approx e'$. TransH [32] improves it by introducing hyperplanes. Bilinear models views KG as 3-order tensor, where each element corresponds to a triple. DistMult [36] adopts $eM_r e'$ to reconstruct the 3-order tensor, TuckER [2] applies Tucker composition [26]. SimplE [12] further considers inverse relations to cover asymmetric features. Neural networks are also utilized, such as ConvE [6] applies 2D convolution on the concatenation of embeddings in each triple. Some studies utilize graph neural networks [21, 27], while they can only apply to KGs merely covering binary knowledge.

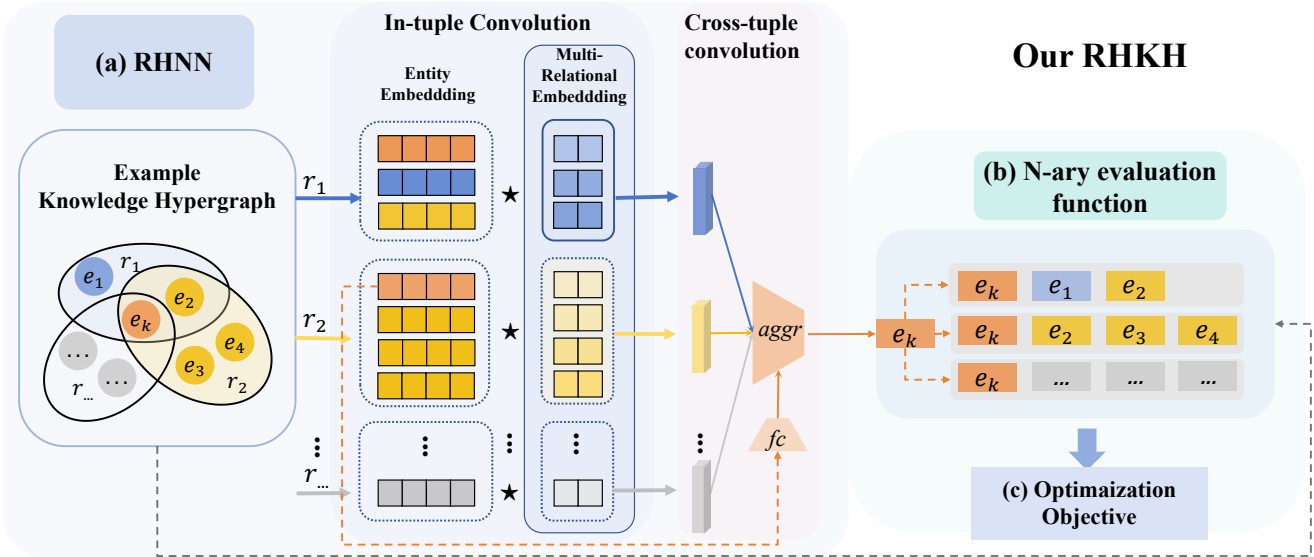

Figure 3: Overview of RHKH. Firstly, RHNN is adopted to consider associations within and among each tuple in a hierarchical way to aggregate and combine entity embeddings with their neighboring features, from entity-relation-interaction at each-arity position, in-tuple convolution to cross-tuple convolution. Then, based on these enhanced entity and relation embeddings, the evaluation function towards N-ary tuple works to calculate confidence score of each tuple. Finally, synthesizing these score, optimization is carried out to data-fit the KH.

## 2.2 Link Prediction on N-ary KHs

Studies devoted to N-ary KHs are significantly scarcer, which generally follow above three categories. For each tuple, translation model m-TransH [33] generalizes TransH [32] via projecting entities in the same tuple onto the relation-specific hyperplane. Its credibility is defined by the weighted sum of projected results. HSimplE and HypE [7] further introduce positional convolutional weight filters to capture the role of entities at different arity-positions. BoxE [1] is a spatio-translational model, inducing entities in the same tuple to be located in a geometric region. The generalized bilinear model GETD [15] is based on Tucker and Tensor ring decomposition [41]. However, it can only apply to KHs whose tuples are with the same arity number.

As neural-network-based models, NaLP [11] and RAM [16] represent each tuple as role-value pairs and applies fully-connected layers to discovery their pairwise relatedness, while HyConvE [29] utilizes both 2D and 3D convolution. Others reconstruct the tuple into a primary triple with additional descriptions $\{(e, r, e'), \{(r_i : v_i)\}_i^n\}$. However, not each tuple has a primary triple [11]. Even ignored this, STARE [9], QUAD [22] and Hy-Transformer [37] completely ignore associations among $\{(r_i : v_i)\}_i^n$. They essentially treat N-ary KHs as KGs with additional information. And they only study link prediction on primary triples, rather than the whole N-ary tuple. Therefore, we do not focus much on them or consider them as baselines. HINGE [19], GRAN [31] and NeuInfer [10] also mainly consider the principal and subordinate relationship, namely the information flow from $\{(r_i : v_i)\}_i^n$ to $(e, r, e')$. Relatedness among $\{r_i : v_i\}_i^n$ and information flow from $(e, r, e')$ to $\{(r_i : v_i)\}_i^n$ is ignored, even though $\{v_i\}_i^n$ are also entities in the original tuple.

Therefore, as aforesaid in Sec.1, existing studies almost all reconstruct original knowledge format in KHs. And the associations within and among tuples are largely ignored by them.

## 3 PROBLEM STATEMENT

KH is denoted as $\mathcal{H} = (\mathcal{E}, \mathcal{R}, \mathcal{K})$, where $\mathcal{E}, \mathcal{R}, \mathcal{K}$ respectively refer to the set of entities, relations and tuples in $\mathcal{H}$. Each **N-ary tuple** $\tau_i \in \mathcal{K}$ represents a piece of knowledge, defined as $\tau_i$ : $r(e_1, e_2, ..., e_{|r|})$, $|r|$ is the arity number of $r$, namely the number of entities that instances of $r$ cover. It is fixed for each $r$, and $\forall r \in \mathcal{R}$, $|r| \geq 2$ in KHs. Assuming that $\mathcal{K}_r$ is the set of tuples with $r$, $\mathcal{K}_r \subseteq \{r\} \times \mathcal{E}^{|r|}$. We use bold lower-case to represent vectors in the embedding space, like $\boldsymbol{e}$ for $e \in \mathcal{E}$. **Arity-position** represents the position of entities in each tuple. For example, $e_1$ is at the $1_{st}$ arity-position of $r_1$ in the tuple $r_1(e_1, e_2, e_3)$.

Formally, let $\mathcal{K}^*$ be the underlying complete knowledge set in the real world, KH completion (KHC) is for predicting the missing knowledge in $\mathcal{K}$, namely $\mathcal{K}^* \ \mathcal{K}$. Link prediction is the common means for implementing KH completion. It discoveries the missing element in a given tuple via existing information in the KH. For example, identifying the most possible candidate for $r(e_1, e_2, ?, e_4)$.

## 4 THE RHKH METHOD

Instead of simplifying full semantics and intrinsic relatedness of each tuple through converting knowledge structure in previous studies, RHKH is based on original knowledge formats in KHs. Challenged by complicated compositions in original N-ary tuples, RHKH adopts our novel relational hypergraph neural network, which could consider associations within and among each tuple.

To refine the association in every tuple, interactions of entities and relation at each arity-position are modeled, along with position-specific correlation.

RHKH is shown in *Figure 3*. Firstly, we employ our RHNN to learn neighborhood information and enhance entity embeddings. Then, N-ConvE is proposed to evaluate the truth value of tuples. Based on it, the optimization objective is defined, enabling RHKH gradually learns semantics indicated by tuples. They are introduced in Sec.4.1, 4.2, 4.3, respectively.

## 4.1 Relational Hypergraph Neural Network

*Figure 4* shows example work-flow of our relational hypergraph neural netowrk, which aggregates and combines neighboring features in KHs. Overall, it follows a hierarchical aggregation scheme that works within and among each neighboring tuple in turn. Different from general hypergraphs, the peculiarity of KHs is that their edges are not links without any meaning, but relations with various semantics. Relations play a significant role, and entities often highlight different aspects under the effect of distinct relations [30]. Therefore, the interaction between entities and relation in each tuple is considered. Semantic components of every relation at each arity-position are also distinguished to refine the characterization of such interaction.

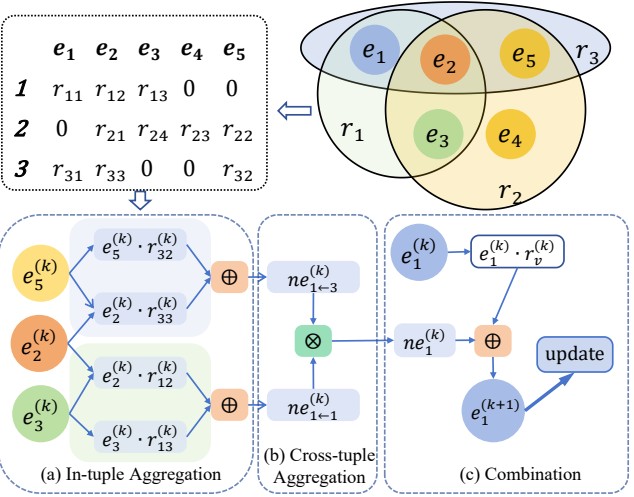

**Figure 4: An example of updating the embedding of $e_1$ through neighboring features $ne_1^{(k)}$ learned by RHNN.**

We first define the neighborhood matrix $\mathcal{M}^{|\mathcal{K}|\times|\mathcal{E}|}$. Given $r_1(e_1, e_2, e_3)$, $r_2(e_2, e_5, e_4, e_3)$ and $r_3(e_1, e_5, e_2)$ in an example KH, $\mathcal{M}$ is shown in *Figure 4*. Each tuple covering $e$ is one of its neighboring tuples, corresponding to a row in $\mathcal{M}$. We use circular correlation [18] to model rich interactions between relation and entities in tuples, it is the compression of tensor product that can capture rich interactions [23]. For each entity-relation pair, it is defined below. $\mathcal{F}(\cdot)$ is fast Fourier transform, $\odot$ is Hadamard product and $\overline{(\cdot)}$ denotes the complex conjugate.

$$e \star r = \mathcal{F}^{-1}(\overline{\mathcal{F}(e)} \odot \mathcal{F}(r)) \tag{1}$$

Compared with binary relations, N-ary ones generally express more complicated knowledge, including multiple sub-semantic components. For example, three components of 3-ary relation *actInFilm* respectively highlight aspects *Actor*, *Role* and *Film* of involved entities at corresponding arity-position. Based on such observation, representing $r$ with a unique vector cannot cover its various semantic components, nor can it express the interaction of $r$ with entities at its different arity-positions properly. To represent the role of various semantic components, we denote $r$ as $\{r, r_1, r_2, ..., r_{|r|}\}$. $r$ is the embedding of $r$ itself, representing its own features. $r_i$ refers to the effect of $r$ on its $i_{th}$ arity-position.

For each entity $e_i$, the neighborhood information obtained from its $m_{th}$ neighboring tuple $\tau_m : r_j(e_1, ..e_{|r_j|})$ is defined in formula(2). As shown in *Figure 4(a)*, neighboring information of $e_1$ from $r_1(e_1, e_2, e_3)$ is $e_2 \star r_{12} + e_3 \star r_{13}$. Similarly, in addition to $e_1$, $r_1(e_1, e_2, e_3)$ also indicates corresponding neighboring information to $e_2$ and $e_3$.

$$ne_{i \leftarrow m} = \sum_{p=1, p \neq i}^{|r_j|} e_p \star r_{jp} \tag{2}$$

Then, from each row in $\mathcal{M}$, corresponding neighboring information to every included entity is obtained. These $ne_{i \leftarrow m}$ are aggregated to constitute neighboring feature of each entity $e_i$. It is defined as formula(3). $N_{e_i}$ is the set of neighboring tuples of $e_i$, $W_1$ is the linear transformation matrix for extracting features and $\sigma(\cdot)$ is activation function, corresponding to *Figure 4(b)*.

$$ne_i = \sigma(\sum_{\tau_m : r_j(e_1, ..e_{|r_j|}) \in N_{e_i}} W_1 \sum_{p=1, p \neq i}^{|r_j|} e_p \star r_{jp}) \tag{3}$$

Next, $e_i$ is combined with $ne_i$ to retain the feature of $e_i$ itself. A virtual relation $r_v$ is introduced, which only represents the links pointing to entities themselves. Thus, it is denoted as a unique vector, $r_v : \{r_v\}$. The entity embedding enhanced with neighboring features is defined as formula(4).

$$e_i = \sigma((\sum_{r_j(e_1, ..e_{|r_j|}) \in N_{e_i}} W_1 \sum_{p=1}^{|r_j|} e_p \star r_{jp}) + W_2(e_i \star r_v)) \tag{4}$$

During model training, in the $(k+1)_{th}$ iteration, $e_i^{(k+1)}$ is obtained through the embeddings in the $k_{th}$ iteration, as shown in formula(5) below, as shown in *Figure 4(c)*.

$$e_i^{(k+1)} = \sigma((\sum_{r_j(e_1, ..e_{|r_j|}) \in N_{e_i}} W_1^{(k)} \sum_{p=1, p \neq i}^{|r_j|} e_p^{(k)} \star r_{jp}^{(k)}) + W_2^{(k)}(e_i^{(k)} \star r_v^{(k)})) \tag{5}$$

Above process could also be expressed as matrix calculation in formula(6). $\mathcal{E}^{(k)} \star \mathcal{R}_v^{(k)} W_2^{(k)}$ combines entity embeddings, as formula(5) shows. The part before it aggregates neighboring features, as shown in formula(2-3). $\star$ is the same as it is defined in formula (1), except that it is applied on matrices here. $\mathcal{E}^{(k)} \in \mathbb{R}^{|\mathcal{E}|\times d}$ is entity embeddings matrix in the $k_{th}$ iteration, $H_r^{(k)} \in \mathbb{R}^{|\mathcal{K}|\times|\mathcal{E}|\times d'}$ acts like neighborhood matrix. For example, $r_1(e_1, e_2, e_4)$ corresponds to the row $[r_{11}^{(k)}, r_{12}^{(k)}, 0, r_{13}^{(k)}...]$ in $H_r^{(k)}$. $\oplus_q$ sums over the

$q_{th}$ dimension of $H_r D_e^{-\frac{1}{2}} \mathcal{E}^{(k)}$, , like $(\oplus_2 (H_r D_v^{-\frac{1}{2}} \mathcal{E}^{(k)})) \in \mathbb{R}^{|\mathcal{K}| \times d}$, $(\oplus_1 (H_r D_v^{-\frac{1}{2}} \mathcal{E}^{(k)})) \in \mathbb{R}^{|\mathcal{E}| \times d}$. $\mathcal{R}_v^{(k)} \in \mathbb{R}^{d \times d}$ is the concatenation of virtual relation $r_v$. $W_1^{(k)}, W_2^{(k)} \in \mathbb{R}^{d \times d'}$ are for feature extraction. $d$ and $d'$ could be the same or different. $H_e^{|\mathcal{E}| \times |\mathcal{K}|}$ is the incidence matrix, $H_e[i, j] = 1$ if $e_i \in \tau_j$. $D_e^{|\mathcal{E}| \times |\mathcal{E}|}$ is the diagonal matrix for normalization, $D_e[i, i] = \sum_{j=1}^{|\mathcal{K}|} H_e[i, j]$.

$$\mathcal{E}^{(k+1)} = \sigma(D_e^{-\frac{1}{2}}(((H_e \oplus_2) - \oplus_1)(H_r^{(k)} \star ($$
$$D_e^{-\frac{1}{2}} \mathcal{E}^{(k)})))W_1^{(k)} + \mathcal{E}^{(k)} \star \mathcal{R}_v^{(k)} W_2^{(k)}) \quad (6)$$

Through such a hierarchical aggregation scheme in RHNN, local context information of entities are encoded into their representations, namely the correlation within and among each high-ary tuple. RHNN also models the interaction of entities and relations at each arity-position meticulously, which is crucial for knowledge reasoning [31].

## 4.2 Evaluation Function

Then, based on above neighborhood-enhanced entity embeddings, the evaluation function on each tuple is defined. It aims to characterize the credibility of correct or false knowledge, thus enabling the model able to gradually distinguish between them during training. Challenged by the complex compositions in N-ary tuples, it utilizes powerful neural network to extract interaction features within each tuple.

We generalize ConvE to propose a N-ConvE, which could be applied to N-ary KHs. In KGs, for each triple $(e, r, e')$, ConvE employs 2D convolution and fully connect layers on the concatenation of $e$ and $r$ to establish their relatedness. Further interacted with target entity $e'$, the overall credibility of the triple is evaluated. It is defined in formula(7). Symbols definitions are the same with those in formula(9) below.

$$f_{r(e, e')} = \sigma(\text{vec}(\sigma([\overline{e}; \overline{r}] * \omega))Y)e' \quad (7)$$

While in KHs, for tuple $r_i(e_1, .., e_{|r_i|})$, N-ConvE discusses its overall credibility via regarding each of the involved entities as the target entity successively. In addition, the role of various semantic components of each relation are also distinguished in evaluation function, thus analyzing the interaction at each arity-position elegantly. Such position-related correlation is also imposed on entity embeddings through position-specific shift. For any $r_i(e_1, .., e_{|r|})$, the shift operator is defined via a linear layer in formula(8). $[v_1; v_2]$ performs concatenation on $v_1$ and $v_2$, $onehot(|r_i|, p)$ is one-hot encoding, returning the $|r_i|$-dimensional vector $v$, $v[p] = 1$, $v_{q \neq p}[q] = 0$. $Z$ is for linear transformation, $b$ is bias vector.

$$\theta_{r_i}(e_p, p) = [e_p; onehot(|r_i|, p)]Z + b \quad (8)$$

Formally, our evaluation function for each N-ary tuple is defined below. Formula (9) characterizes the credibility of $r_i(e_1, .., e_{|r_i|})$ when regarding $e_p$ in it as the target entity, it is the extended form of formula(7) for tuples. The whole evaluation function is in formula (10). $\circ$ denotes Hadamard production, establishing the interactions between corresponding semantic component of the relation and the entity at each arity-position. $\overline{(v)}$ reshapes $v^{k \times 1}$, transforming it into $v^{k_w \times k_h}$, $k = k_w k_h$. $\sigma$ is activation function, $*$ refers to

**Table 1: Statistics of the datasets.**

| | $|\mathcal{E}|$ | $|\mathcal{R}|$ | $|\mathcal{K}_{train}|$ | $|\mathcal{K}_{valid}|$ | $|\mathcal{K}_{test}|$ |
|---|---|---|---|---|---|
| FB-AUTO | 3410 | 8 | 6778 | 2255 | 2180 |
| JF17K | 29177 | 327 | 61911 | 15822 | 24915 |
| M-FB15K | 10314 | 71 | 415375 | 39348 | 38797 |
| WN18RR | 40943 | 11 | 86835 | 3034 | 3134 |
| FB15K-237 | 14541 | 237 | 272115 | 17535 | 20466 |

convolution filters, $vec(\cdot)$ flattens the tensor into a vector. $Y$ is for linear transformation. $\theta_{r_i}$ is the shift operator defined in formula(8), acting on entities, $[\cdot; \cdot]$ is concatenation operator.

$$f_{r_i(e_1, e_2, .., e_{|r_i|})}^p = \sigma(\text{vec}(\sigma([\overline{\sum_{j=1, j \neq p}^{|r_i|} (\theta_{r_i}(e_j \circ r_{ij}, j)); \overline{r_i}]}$$
$$* \omega))Y)(\theta_{r_i}(e_p \circ r_{ip}, p)) \quad (9)$$

$$f_{r_i(e_1, e_2, .., e_{|r_i|})} = \frac{1}{|r_i|} \sum_{p=1}^{|r_i|} f_{r_i(e_1, e_2, .., e_{|r_i|})}^p \quad (10)$$

We employ such a neural-network-based evaluation function due to the satisfactory ability of neural networks in KGs [24]. For comparison, we also generalize the representative evaluation functions of other two categories, namely translation-based TransE and bilinear DistMult. However, they are not expressive enough to evaluate N-ary knowledge, which could be demonstrated in experiments in Sec.5.2. N-TransE and N-DistMult are defined below.

$$f'_{r_i(e_1, e_2, .., e_{|r_i|})} = (\sum_{p=1}^{|r_i|} shift_{r_i}(e_p \circ r_{ip}, p)) + r_i \quad (11)$$

$$f''_{r_i(e_1, e_2, .., e_{|r_i|})} = (\prod_{p=1}^{|r_i|} shift_{r_i}(e_p \circ r_{ip}, p))r_i \quad (12)$$

## 4.3 Optimization Objective

Based on above process, the whole optimization objective of RHKH is defined in formula(13). Overall, the goal of training is to make the embedding space gradually conform to semantics indicated by tuples in the KH, thus enabling the model to distinguish between right and wrong knowledge tuples. Towards such target, we define the cross-entropy loss function to establish score gap between correct and wrong tuples as follows. And the optimization process is carried out using Adam optimizer [13]. Tuples in the KH naturally constitutes positive samples. Negative samples of $\tau_m : r_i\{e_1, ..., e_{r|i|}\} \in \mathcal{K}$ is constructed through replacing $e_{p:p \in \{1, 2, ..., |r_i|\}}$ in $\tau_m$ with other $e' \in \mathcal{E}$ in turn, denoted as $\tau'_m$. $\mathcal{K}_{\tau_m}$ is the set of negative instances corresponding to $\tau_m$, $\mathcal{K}_{\tau_m}^-$, plus $\tau_m$. $f$ is the evaluation function defined in formula(10). $y$ refers to 0,1 label, positive tuples are marked with 1 and negative ones are

---

**Algorithm 1** RHKH

---

**Require:** KH $\mathcal{H} = (\mathcal{E}, \mathcal{R}, \mathcal{K})$, maximum Iteration $N$, batch size

1: Initialization
2: **for** k = 1 **to** $N$ **do**
3:     Aggregate neighboring features, as shown in formula (2-3).
        $\mathcal{NE}^{(k)} \leftarrow aggregate(\mathcal{E}^{(k)}, \mathcal{R}^{(k)}, \mathcal{M})$
4:     Combine neighboring features, as shown in formula (4).
        $\mathcal{E}^{(k+1)} \leftarrow combine(\mathcal{NE}^{(k)}, \mathcal{N}^{(k)})$
5:     **for each** batch $S \subseteq \mathcal{K}$ **do**
6:         Negative sampling as required in formula (13)
            $S^+ = S \cup \{\mathcal{K}_{\tau_m}^- | \tau_m \in S\}$
7:         Update embeddings w.r.t $\nabla L$ in formula (13).
8:     **end for**
9: **end for**

---

with 0. $s(\cdot)$ is the logistic sigmoid function.

$$L = \sum_{\tau_m \in \mathcal{K}} (-\frac{1}{|\mathcal{K}_{\tau_m}|} \sum_{\tau_{m'} \in \mathcal{K}_{\tau_m}} (y_{\tau_{m'}} \cdot \log(s(f_{\tau_{m'}}))$$
$$+ (1 - y_{\tau_{m'}}) \cdot \log(1 - s(f_{\tau_{m'}})))) \qquad (13)$$

The workflow of RHKH is summarized in Algorithm 1. RHKH first initializes parameters and embeddings of entities, relations in the KH, as Line 1 shows. As the iteration shown in Line 2-8, associations within and among N-ary tuples are first modeled through aggregating neighboring features of entities, as shown in formula(2-3). Then, aggregated features are combined with the embeddings of entities themselves, as discussed in formula(4). Finally, as Line 5-8 shows, RHKH negatively samples each training batch and performs optimization for current batch.

## 5 EXPERIMENTS

The performance of RHKH is evaluated in this section. We first introduce our experimental setup. Then, the experiments results for each task are reported in turn.

### 5.1 Experimental Setup

*5.1.1 Evaluation Protocol.* For each test tuple $\tau_m : r_i(e_1, ..., e_{|r_i|}) \in \mathcal{K}_{test}$, entities at each of its arity-position are taken into evaluation. Specifically, $\forall e \in \mathcal{E}$ will be filled into the position being evaluated, $p$, in turn. Then, these constituted tuples $r_i(e_1, ..., e, ...e_{|r_i|})$ are rated via evaluation function in Sec 4.2. The scores are sorted, where the rank of the correct answer $e_p$ is naturally available. Performing such process at each arity-position for every tuple in the test set, the value of evaluation metrics *MRR* and *Hit@K* could be obtained.

*5.1.2 Datasets.* Following previous work, the experiments are conducted on three N-ary KH datasets, namely JF17K, FB-AUTO and M-FB15K [7]. When testing RHKH on different arity number, 2-ary KGs are also considered, including classical FB15K-237 [25] and WN18RR [6]. *Table 1* shows statistics.

*5.1.3 Baselines.* Existing work for link prediction on N-ary KHs constitute the major comparison, including m-TransH [33], BoxE [1], NaLP [11], NeuInfer [10], GRAN [31], HINGE [19], GETD [15], RAM [16], HyConvE [29], along with HypE, HSimplE, m-CP and m-DistMult in [7]. GETD could only be applied to the KHs whose

tuples are all with the same number of arity. Therefore, we split each dataset according to arity number of tuples and run GETD on the decomposed datasets in turn. GETD reports the synthesis of results on these decomposed datasets.

*5.1.4 Implementations.* We employ $\tanh(\cdot)$ as activation function $\sigma$ in RHKH. The optimal combination of hyperparameters is determined via the performance on valid set. The dimension of embeddings is set to 200, batch size is 256, and maximum iteration is 500. The learning rate $lr$ is in $\{0.0005, 0.001, 0.005, 0.01\}$. Dropout rate in and after RHNN $hd, d$, feature dropout $fd$ in N-ConvE are all in $\{0.0, 0.1, 0.2, 0.3\}$. Specifically, on experiments datasets, $lr = 0.001, hd = 0.0, d = 0.2, fd = 0.3$, besides $fd = 0.2$ on JF17K. RHKH is implemented using V100. For the baselines, we follow their provided codes and optimal settings to run them in our experimental environment and report corresponding results. The code of RHKH and datasets are provided as the supplementary material, and hyperparameters value on each dataset are also annotated.

### 5.2 Experiments Results

*5.2.1 Link Prediction on N-ary KHs.* We first evaluate the reasoning performance of RHKH on KHs via link prediction task. *Table 2* shows the results. In addition to primary baselines introduced in Sec.5.1, the scheme converting KHs into triples and then applying models for link prediction on KGs is also considered. Actually, almost all previous studies including [7, 11, 33] prove the defects of such scheme theoretically and experimentally, such as information loss and poor performance. Even though, we still include this idea into comparison and apply several representative models for KGs here to fully demonstrate the effectiveness of RHKH.

With similar thought, [7] employs r-SimplE as the baseline, which uses SimplE on datasets transformed through reification. Following such approach, we also first transform KHs into triples via reification or S2C, and then apply typical models proposed for KGs. Specifically, TransE, RotatE, SimplE and CompGCN, which are respectively the representative of three categories, namely translation methods, bilinear models and neural-network-based ones. In *Table 2*, $s-$ and $r-$ denote that the model runs on the datasets transformed through S2C or reification. For all baselines, their results are obtained by following their provided codes and optimal settings in our experimental environment.

As *Table 2* shows, by comparison, applying models for KGs on the transformed KHs performs generally poor. Such conclusion is consistent with previous studies like [7], further confirming the necessity of specifically proposing a model for KHs. Furthermore, our RHKH outperforms other baselines and shows satisfactory performance. This demonstrates the effectiveness of our model.

*5.2.2 Study on Instances Datasets of Different Arity Number.* We further analyze the performance of RHKH on various arity numbers. Each dataset is divided according to the arity number $|r_i|$ of tuples $r_i(e_1, ..., e_{|r_i|})$. Compared with the baselines proposed for KHs, results are shown in *Table 3*. For the arity number two, in *Table 4*, we additionally consider KGs since they only cover 2-ary tuples, namely WN18RR and FB15K-237. It is also for demonstrating the

Table 2: Experimental results of link prediction on N-ary KHs.

| | FB-AUTO | | | | JF17K | | | | M-FB15K | | | |
| --- | --- | --- | --- | --- | --- | --- | --- | --- | --- | --- | --- | --- |
| | MRR | Hit@10 | Hit@3 | Hit@1 | MRR | Hit@10 | Hit@3 | Hit@1 | MRR | Hit@10 | Hit@3 | Hit@1 |
| r-TransE | 0.249 | 0.509 | 0.291 | 0.132 | 0.234 | 0.432 | 0.274 | 0.133 | 0.122 | 0.244 | 0.126 | 0.060 |
| s-TransE | 0.134 | 0.380 | 0.183 | 0.021 | 0.114 | 0.345 | 0.112 | 0.018 | 0.085 | 0.228 | 0.062 | 0.027 |
| r-RotatE | 0.338 | 0.551 | 0.390 | 0.221 | 0.391 | 0.603 | 0.442 | 0.280 | 0.176 | 0.391 | 0.182 | 0.081 |
| s-RotatE | 0.160 | 0.395 | 0.225 | 0.038 | 0.145 | 0.399 | 0.145 | 0.040 | 0.071 | 0.178 | 0.043 | 0.023 |
| r-SimplE | 0.266 | 0.444 | 0.321 | 0.169 | 0.126 | 0.285 | 0.129 | 0.056 | 0.052 | 0.112 | 0.062 | 0.016 |
| s-SimplE | 0.161 | 0.372 | 0.216 | 0.050 | 0.104 | 0.253 | 0.097 | 0.035 | 0.085 | 0.209 | 0.074 | 0.027 |
| r-CompGCN | 0.349 | 0.364 | 0.349 | 0.342 | 0.416 | 0.440 | 0.415 | 0.404 | 0.345 | 0.355 | 0.344 | 0.341 |
| s-CompGCN | 0.454 | 0.602 | 0.485 | 0.380 | 0.395 | 0.582 | 0.430 | 0.301 | 0.398 | 0.474 | 0.413 | 0.359 |
| m-TransH | 0.788 | 0.869 | 0.824 | 0.740 | 0.422 | 0.570 | 0.466 | 0.336 | 0.614 | 0.808 | 0.661 | 0.518 |
| m-DistMult | 0.776 | 0.856 | 0.826 | 0.720 | 0.454 | 0.635 | 0.503 | 0.359 | 0.708 | 0.851 | 0.745 | 0.634 |
| m-CP | 0.780 | 0.838 | 0.804 | 0.747 | 0.402 | 0.564 | 0.446 | 0.317 | 0.676 | 0.827 | 0.713 | 0.599 |
| HypE | 0.805 | 0.846 | 0.825 | 0.779 | 0.496 | 0.658 | 0.542 | 0.409 | 0.764 | 0.870 | 0.788 | 0.711 |
| HSimplE | 0.758 | 0.858 | 0.805 | 0.698 | 0.474 | 0.654 | 0.526 | 0.376 | 0.738 | 0.863 | 0.769 | 0.674 |
| BoxE | 0.826 | 0.893 | 0.847 | 0.790 | 0.553 | 0.718 | 0.599 | 0.465 | 0.755 | 0.875 | 0.785 | 0.695 |
| NaLP | 0.634 | 0.755 | 0.683 | 0.563 | 0.202 | 0.294 | 0.216 | 0.152 | 0.104 | 0.170 | 0.106 | 0.067 |
| GRAN | 0.768 | 0.903 | 0.860 | 0.670 | 0.533 | 0.716 | 0.581 | 0.438 | 0.538 | 0.788 | 0.610 | 0.408 |
| GETD | 0.751 | 0.830 | 0.773 | 0.715 | 0.561 | 0.703 | 0.593 | 0.482 | 0.701 | 0.816 | 0.726 | 0.641 |
| NeuInfer | 0.622 | 0.706 | 0.659 | 0.569 | 0.340 | 0.454 | 0.369 | 0.282 | 0.348 | 0.554 | 0.379 | 0.252 |
| HINGE | 0.588 | 0.700 | 0.633 | 0.520 | 0.416 | 0.575 | 0.449 | 0.334 | 0.169 | 0.293 | 0.184 | 0.106 |
| RAM | 0.803 | 0.879 | 0.834 | 0.759 | 0.548 | 0.698 | 0.582 | 0.471 | 0.761 | 0.871 | 0.785 | 0.706 |
| HyConvE | 0.764 | 0.847 | 0.792 | 0.718 | 0.534 | 0.690 | 0.570 | 0.453 | 0.438 | 0.636 | 0.468 | 0.343 |
| RHKH | 0.877 | 0.913 | 0.890 | 0.857 | 0.565 | 0.713 | 0.602 | 0.487 | 0.808 | 0.905 | 0.827 | 0.765 |

Table 3: *MRR* of link prediction on tuples with different arity number.

| | FB-AUTO | | | | JF17K | | | | M-FB15K | | | |
| --- | --- | --- | --- | --- | --- | --- | --- | --- | --- | --- | --- | --- |
| | Arity=2 | 3 | 4 | ≥5 | Arity=2 | 3 | 4 | ≥5 | Arity=2 | 3 | 4 | ≥5 |
| m-TransH | 0.370 | - | 0.285 | 0.895 | 0.213 | 0.452 | 0.622 | 0.764 | 0.515 | 0.609 | 0.833 | 1.000 |
| m-DistMult | 0.501 | - | 0.301 | 0.849 | 0.316 | 0.486 | 0.585 | 0.600 | 0.526 | 0.720 | 0.875 | 0.964 |
| m-CP | 0.435 | - | 0.134 | 0.874 | 0.241 | 0.452 | 0.492 | 0.628 | 0.522 | 0.682 | 0.938 | 0.995 |
| HypE | 0.411 | - | 0.344 | 0.905 | 0.291 | 0.536 | 0.694 | 0.756 | 0.687 | 0.764 | 1.000 | 1.000 |
| HSimplE | 0.499 | - | 0.330 | 0.827 | 0.318 | 0.531 | 0.569 | 0.600 | 0.587 | 0.747 | 0.875 | 0.999 |
| BoxE | 0.458 | - | 0.447 | 0.919 | 0.365 | 0.568 | 0.794 | 0.817 | 0.582 | 0.767 | 1.000 | 1.000 |
| NaLP | 0.038 | - | 0.043 | 0.782 | 0.053 | 0.172 | 0.551 | 0.336 | 0.068 | 0.080 | 0.005 | 0.712 |
| GRAN | 0.362 | - | 0.524 | 0.864 | 0.283 | 0.578 | 0.788 | 0.847 | 0.388 | 0.547 | 0.778 | 0.791 |
| GETD | 0.533 | - | 0.224 | 0.813 | 0.329 | 0.620 | 0.750 | 0.829 | 0.697 | 0.709 | 0.875 | 0.540 |
| NeuInfer | 0.134 | - | 0.115 | 0.743 | 0.122 | 0.380 | 0.567 | 0.587 | 0.137 | 0.393 | 0.001 | 0.009 |
| HINGE | 0.073 | - | 0.222 | 0.713 | 0.216 | 0.441 | 0.722 | 0.508 | 0.212 | 0.143 | 0.001 | 0.588 |
| RAM | 0.529 | - | 0.449 | 0.873 | 0.334 | 0.590 | 0.732 | 0.830 | 0.624 | 0.768 | 0.877 | 1.000 |
| HyConvE | 0.198 | - | 0.457 | 0.897 | 0.287 | 0.587 | 0.768 | 0.792 | 0.408 | 0.415 | 1.000 | 1.000 |
| RHKH | 0.545 | - | 0.463 | 0.962 | 0.314 | 0.624 | 0.774 | 0.887 | 0.724 | 0.810 | 0.875 | 1.000 |

effectiveness of our model against traditional KGs. The representative models only proposed for KGs are also considered as baselines in *Table 4*, their results are from [17].

From *Table 3*, RHKH almost outperforms all baselines, including tuples of various arity number and 2-ary triples. This point is also indicated by *Table 4*. When applied to KGs, RHKH shows obvious advantages over other models proposed for N-ary KHs. Meanwhile, it is also competitive with the work put forward only for KGs.

These conclusions show the applicability of RHKH in the face of knowledge with different arity number, and further demonstrate its effectiveness.

5.2.3 *Ablation Study*. Ablation study is carried out to test the effectiveness of each major component in RHKH. Specifically, $-multi$ no longer distinguishes various semantics components of each relation and $-shift$ omits the position-specific shift. $-r$ regards KHs

**Table 4: *MRR* and *Hit@1* of link prediction on 2-ary KGs.**

| | WN18RR | | FB15K-237 | |
|---|---|---|---|---|
| | MRR | Hit@1 | MRR | Hit@1 |
| **TransE** | 0.226 | 0.056 | 0.294 | 0.198 |
| **RotatE** | 0.476 | 0.428 | 0.337 | 0.241 |
| **DistMult** | 0.430 | 0.390 | 0.241 | 0.155 |
| **TuckER** | 0.470 | 0.443 | 0.358 | 0.266 |
| **ConvE** | 0.430 | 0.400 | 0.325 | 0.237 |
| **CompGCN** | 0.479 | 0.443 | **0.355** | 0.264 |
| **m-TransH** | 0.310 | 0.267 | 0.182 | 0.110 |
| **m-DistMult** | 0.399 | 0.381 | 0.231 | 0.142 |
| **m-CP** | 0.026 | 0.017 | 0.222 | 0.140 |
| **HypE** | 0.380 | 0.359 | 0.268 | 0.170 |
| **HSimplE** | 0.181 | 0.136 | 0.252 | 0.161 |
| **BoxE** | 0.441 | 0.395 | 0.320 | 0.220 |
| **NaLP** | 0.394 | 0.371 | 0.160 | 0.119 |
| **GRAN** | 0.452 | 0.418 | 0.342 | 0.251 |
| **GETD** | 0.437 | 0.406 | 0.300 | 0.217 |
| **NeuInfer** | 0.051 | 0.044 | 0.136 | 0.093 |
| **HINGE** | 0.311 | 0.263 | 0.184 | 0.120 |
| **RAM** | 0.425 | 0.361 | 0.266 | 0.183 |
| **HyConvE** | 0.284 | 0.200 | 0.256 | 0.163 |
| **RHKH** | **0.480** | **0.445** | 0.353 | **0.271** |

**Table 5: *MRR* on ablated models.**

| | FB-AUTO | JF17K | M-FB15K |
|---|---|---|---|
| **-multi** | 0.861 | 0.547 | 0.791 |
| **-shift** | 0.860 | 0.557 | 0.795 |
| **-r** | 0.858 | 0.553 | 0.796 |
| **-/+translation** | 0.837 | 0.536 | 0.659 |
| **-/+bilinear** | 0.855 | 0.535 | 0.727 |

as general hypergraphs like previous HGNN do, various relation semantics on hyperedges are ignored. That is, neighborhood aggregation only works on neighboring entities, the effect of relations are completely removed in formula (2-5) in RHNN. $-/+translation$ and $-/+bilinear$ refer to replacing our evaluation function with N-TransE and N-DistMult in formula (11-12), respectively. Results are reported in *Table 5*.

As *Table 5* shows, omitting each of these major components affects the effectiveness of RHKH, demonstrating their role in RHKH. The comparison with $-/+translation$ and $-/+bilinear$ reflects that relatively simple translation and bilinear evaluation function may be insufficient to capture intrinsic correlation in N-ary tuples, stating our motivation for adopting a neural-network-based evaluation function.

To sum up, firstly, since RHKH is based on original knowledge format in KHs, possible efforts of converting N-ary tuples and picking primary triples in previous work could be saved. Secondly, compared with existing studies, RHKH almost always shows advantages, including KHs and KG, demonstrating its effectiveness and adaptability for various arity number. Thirdly, in ablation study,

omitting each of its major components affects RHKH, embodying their role.

## 6 CONCLUSION AND FUTURE WORK

Different from previous work, we do not transform the structure of N-ary tuples in KHs into triples or role-value sets, which would cause potential issues like the loss of structural information, the severing of intrinsic relatedness within each tuple and so on. Based on original knowledge format in KHs, we propose the first model for link prediction on KHs based on a novel relational hypergraph neural network (RHNN) specifically proposed for KHs, RHKH. Challenged by complicated compositions indicated by original knowledge formats in KHs, the association within and among each tuple is discovered via hierarchical aggregation in RHNN.

In particular, hyperedges among nodes are with various semantics, rather than the ordinary edges without special meanings is one of the distinguishing features of KHs from general hypergraphs. Towards this, RHKH emphasizes the interaction of relation and entities involved in the same tuple to combine specific relational contexts. To further refine such entity-relation interaction, semantic components at each arity-position of every relation are distinguished, position-specific shift is also introduced. In addition, RHKH proposes evaluation function towards the credibility of N-ary tuples, N-ConvE. Extensive experiments demonstrate the effectiveness and superiority of RHKH.

Regarding future work, various relation patterns in KHs are considered to be discovered and studied.

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
