# OpenReview forum: "RHKH: Relational Hypergraph Neural Network for Link Prediction on N-ary Knowledge Hypergraph"
_acmmm.org/ACMMM/2024/Conference — MM2024 Poster_

### Official Review · Reviewer_x1sR · 2024-05-08

**Rating:** 4
**Confidence:** 3

**Summary:**

This paper proposes the first KH reasoning model based on an innovative relational hypergraph neural network (RHNN), RHKH. Challenged by complicated compositions indicated by the original format of N-ary tuples, association within and among each knowledge is discovered through RHNN. It considers complex interactions between relation and entities involved in the same knowledge as well. Semantic components at each arity-position of relations are distinguished to refine such interactions, along with introducing position-specific shift. Extensive experiments demonstrate the effectiveness of RHKH.

**Strengths:**

There are several innovations in this paper:

(1)  Based on original knowledge formats, the author propose the first model for link prediction on KHs based on a novel relational hyper-graph neural network (RHNN), rather than their simplifying or converting of knowledge structure. The usual preprocessing of converting N-ary tuples and picking primary triples in previous studies is omitted in RHKH, possible manual efforts are also saved.

(2)  Challenged by complicated compositions indicated by original knowledge formats in KHs, association within and among tuples is considered through RHNN.

(3) Evaluation methods on the credibility of each tuple comprehensively considers its inherent relatedness, including the interaction of involved entities and relations at each arity-position and position-related correlations.

(4)  In the experimental section, the authors compare PHPQ with 13 common baselines,so that we can see that the experimental results are the most advanced.

**Limitations:**

Although the paper is well, clearly organized, and logical, some parts must be changed and complemented.

(1)	The font in figure 2 is too small to read clearly.

(2)	In formula (3), what does  ne_{i \to m}, e_p, r_{ip} represent? The first appearance of a letter in a formula requires an explanation.

(3)	What does Y in formula (7) represent?

(4)	What does   shift_{r_i} represent in formulas (11) and (12)?

(5)	Why is only the experimental indicator MRR reported in Table 3?

(6)	Why is only Hit@1 reported in Table 4, but not Hit@3 and Hit@10?

(7)	Why only use MRR as an evaluation indicator in ablation study?

(8)	The references are too old, it is suggested to add two years of literature.

**Suitability:**

2

---

### Official Review · Reviewer_ogsn · 2024-05-26

**Rating:** 4
**Confidence:** 2

**Summary:**

This paper proposes a relational hypergraph neural network for N-ary knowledge hypergraph link prediction. The proposed method RHKH is based on the original knowledge formats in knowledge hypergraph. Experiments shows that RHKH performs well on hypergraph link prediction.

**Strengths:**

1. The motivation of this work is clear, i.e. to utilize the original representation of the knowledge hypergraph without format transforming,  and the proposed method matches the motivation.
2. The paper is clearly presented and easy to follow.
3. The performance of the proposed method performs well on knowledge hypergraph link prediction.

**Limitations:**

1. Some methods mentioned in the Related Work is not compared in the experiments, such as STARE and Hy-Transformer. And why the baselines are chose is not explained.
2. In deep analysis of the model are expected. Apart from the overall experiments, analysis about the benefits of keeping the original format of the hypergraph is expected, through case study or other studies.

**Suitability:**

2

---

### Official Review · Reviewer_8rPJ · 2024-05-27

**Rating:** 5
**Confidence:** 2

**Summary:**

This paper focuses on the knowledge hypergraph (KH) link prediction task. The authors have studied the characteristics of KH information and structure, and proposed a relational hypergraph network for modeling KH's information and semantics. Specifically, the designed method involves aggregation and propagation of information of entities, and a batched training process is adopted for optimizing the model. The authors conduct experiments on three commonly used benchmark datasets, and the results show that the proposed method achieves better performance than the previous methods. With different arity settings, the proposed RHKH still demonstrates stable improvement.

**Strengths:**

1. The formal description is very clear, and the paper is well-written.
2. The experimental results are significant and stable.

**Limitations:**

1. Compared to HyperE, this paper does not introduce particularly impressive technical innovations or new concepts, but rather focuses on model architecture and performance improvement.
2. Considering the used datasets, this paper cannot be regarded as "multimedia" or "cross-modal" in nature.

There are no other significant disadvantages in this paper.

**Suitability:**

2

---

### Official Review · Reviewer_m7mg · 2024-05-29

**Rating:** 5
**Confidence:** 3

**Summary:**

This work proposes a relational hypergraph neural network for link prediction on N-ary knowledge hypergraphs. Different from the previous works of converting N-ary knowledge into triples with auxiliary relations, the proposed method learns the internal associations of N-ary knowledge and distinguishes the relation semantic components at different arity-positions based on the original format of knowledge hypergraph. Extensive experiments have also been done to evaluate the performance of the proposed method on N-ary KH datasets as well as 2-ary KG.

**Strengths:**

The work proposes a novel method for the linkpredictions of N-ary knowledge hypergraphs without reconstructing the original knowledge format. This paper provides a clear problem statement and methodology description. Extensive experiments have been done and compared with adequate baselines to prove the effectiveness of the proposed method.

**Limitations:**

A few layout issues should be modified: texts in line 395 and line 420 exceed the column width. It would be better to add the reference next to each baseline in Table 2, 3 and 4. More results analysis could be added in the experiment result section, e.g. how BoxE achieves much higher performance than the proposed work in JF17k with arity=2?

**Suitability:**

2

---

### Meta-Review · Area_Chair_sv2f · 2024-07-03

**Recommendation:** Accept (Poster)
**Confidence:** 4

**Metareview:**

This paper presents a novel and well-motivated approach for link prediction in N-ary knowledge hypergraphs using a relational hyper-graph neural network (RHNN) without transforming the original knowledge format. The paper is commended for its clear problem statement, methodology, and extensive experimental validation against numerous baselines, highlighting its effectiveness and innovation. However, there are some areas for improvement, including minor layout issues, deeper analysis of results, comparison with additional methods, and clarification of certain formula elements and evaluation metrics.